# Vulvodynia—It Is Time to Accept a New Understanding from a Neurobiological Perspective

**DOI:** 10.3390/ijerph18126639

**Published:** 2021-06-21

**Authors:** Rafael Torres-Cueco, Francisco Nohales-Alfonso

**Affiliations:** 1Department of Physiotherapy, University of Valencia, 46010 Valencia, Spain; 2Gynecology Section, Clinical Area of Women’s Diseases, La Fe University Hospital, 46010 Valencia, Spain; fnohalesa@gmail.com

**Keywords:** vulvodynia, complex pain, central sensitization pain, cognitive-behavioral therapy

## Abstract

Vulvodynia is one the most common causes of pain during sexual intercourse in premenopausal women. The burden of vulvodynia in a woman’s life can be devastating due to its consequences in the couple’s sexuality and intimacy, in activities of daily living, and psychological well-being. In recent decades, there has been considerable progress in the understanding of vulvar pain. The most significant change has been the differentiation of vulvar pain secondary to pathology or disease from vulvodynia. However, although it is currently proposed that vulvodynia should be considered as a primary chronic pain condition and, therefore, without an obvious identifiable cause, it is still believed that different inflammatory, genetic, hormonal, muscular factors, etc. may be involved in its development. Advances in pain neuroscience and the central sensitization paradigm have led to a new approach to vulvodynia from a neurobiological perspective. It is proposed that vulvodynia should be understood as complex pain without relevant nociception. Different clinical identifiers of vulvodynia are presented from a neurobiological and psychosocial perspective. In this case, strategies to modulate altered central pain processing is necessary, changing the patient’s erroneous cognitions about their pain, and also reducing fear avoidance-behaviors and the disability of the patient.

## 1. Introduction

Vulvar pain without a clearly evident pathology has always existed [1]. However, this condition has been and continues to be poorly recognized by health professionals. Until about 40 years ago there were no publications that described this condition. Since then, there has been an exponential increase in its prevalence, surely because of its personal connotations, was not reported by women and also due to greater recognition by health professionals [1].

Vulvodynia is the most common cause of pain during sexual intercourse in premenopausal women [2]. Although no epidemiological study of prevalence has currently been carried out worldwide, it is estimated that vulvodynia affects 8–10% of women of all ages [3].

In an epidemiological study carried out in the United States, they found that up to 16% of women experience vulvodynia throughout their lives [4] and that by age of 40, 7% to 8% of women would have experienced symptoms of vulvodynia at some point in their life [5]. In Spain, a study published in 2019 has shown that the prevalence throughout life reaches 13% of all women [6].

The description of idiopathic vulvar pain, according to Moyal-Barranco and Lynch [1], does not appear until the late 19th century in a book in which Thomas described women with “excessive hypersensitivity of the nerves supplying the mucous membrane of a part of the vulva.” [7].

It is in 1976 at the 3rd *International Society for the Study of Vulvovaginal Disease* (ISSVD) World Congress, when the first modern description of idiopathic vulvar pain is accepted and the term *burning vulva* is adopted [1]. In 1978, Dodson and Friedrich [8] described this same clinical condition in which the women reported pain and dyspareunia but in which no obvious clinical findings were observed, interpreting that it was a psychogenic pain and called it *psychosomatic vulvovaginitis*.

In 1987 Friedrich described the *vulvar vestibulitis syndrome* and its characteristic triad: severe pain on palpation of the vulvar vestibule or when attempted vaginal entry, localized pressure tenderness, and erythema of varying degrees in the vulvar vestibule [9]. In later years, a classification for idiopathic vulvar pain began to be used, differentiating it into 2 types; vestibulitis and *dysesthetic vulvodynia* [10]. Vestibulitis was defined when the Friedrich criteria were met and dysesthetic vulvodynia when the vulvar pain was spontaneous and generalized and in the absence of physical findings. In cases of vestibulitis, vestibulectomy was proposed and pharmacological treatment in dysesthetic vulvodynia.

However, in the late 1990s, disagreement began to develop regarding the reliability and accuracy of these diagnostic labels. Hence, the ISSVD asked 2 members, Micheline Moyal-Barranco and Peter Lynch to develop a new proposal on taxonomy. Finally, at the ISSVD World Congress in 2003, the term vestibulitis was eliminated as it was not possible to confirm inflammatory pathogenesis. In this congress, the term vulvar dysesthesia was also removed because it suggested an underlying neurological aetiology, that has never been demonstrated, and it was decided to use the term vulvodynia [1]. Vulvodynia was then defined as vulvar discomfort, most often described as burning pain, occurring in the absence of relevant visible findings or a specific, clinically identifiable neurological disorder [1].

Moreover, in 2015 it was held a consensus meeting on the taxonomy of pelvic pain between the International Society for the Study of Vulvovaginal Disease (ISSVD), the International Society for the Study of Women’s Sexual Health (ISSWSH), the International Pelvic Pain Society (IPPS) and representatives of the American Congress of Obstetricians and Gynecologists (ACOG), of the American Society for Colposcopy and Cervical Pathology (ASCCP), and of the National Vulvodynia Association (NVA). At this meeting, a new taxonomy was approved and co-published in the following 3 journals: Obstetrics and Gynecology [11], The Journal of Sexual Medicine [12], and The Journal of Lower Genital Tract Disease [13].

The most relevant change was to differentiate *vulvar pain secondary to a specific disorder* such as recurrent candidiasis, postherpetic neuralgia, vulvovaginal atrophy, etc. from *vulvodynia*, in which there is no clear identifiable cause (Table 1). In this new taxonomy, vulvodynia is defined as vulvar pain lasting at least 3 months, without a clear identifiable cause, which may have potential associated factors. An important difference between the new terminology used in the 2015 taxonomy compared to that of 2003 is the addition of the potential associated factors. This addition implies a paradigm shift derived from research showing that some factors may be associated with the development and perpetuation of this clinical condition so that vulvodynia begins to be considered a multifactorial process [12] (Table 2).

These associated factors include many different symptoms and frequent comorbidity in the pelvic area such as urological or coloproctological pain syndromes, pain syndrome associated with endometriosis, and irritable bowel syndrome, but also in remote areas such as orofacial pain or fibromyalgia [14,15,16,17,18,19]. The association of these factors with vulvodynia suggests that this entity is the expression of similar underlying pathophysiological processes [3]. The inclusion of psychosocial factors also seems adequate since the association between pelvic pain and anxiety, depression, post-traumatic stress, sexual problems has been demonstrated by many different studies [20,21,22,23]. The current problem is that this is an ‘agnostic’ taxonomy. Although it differentiates between secondary vulvar pain and vulvodynia as pain when a cause is not evident, it has been included other associated factors: genetic, hormonal, inflammatory, musculoskeletal, and neurological. Therefore, it seems that this proposal wants to indicate that, although a cause of vulvodynia is not detected, it surely will have it. In this way, it is proposed a step-by-step therapeutic approach addressing pelvic floor dysfunction and psychosexual health, together with medical management [24,25,26]. Some authors propose a 3 steps approach [27]. The first step combines vulvar care measures, dietary recommendations, topical and oral medications, pelvic floor muscle training/physiotherapy, and cognitive-behavioral and psychosexual therapy. The second step includes minimally invasive neuromodulation techniques such as multilevel infiltration, radiofrequency, and neurostimulation procedures (infiltration of impar ganglion, botulinum toxin, spinal cord stimulation, selective stimulation of sacral nerve roots). The third therapeutic step proposed is surgery such as vestibulectomy and surgery for pudendal nerve entrapment.

However, currently, there is insufficient evidence of the efficacy of many of these interventions [28]. For example, a placebo was shown to be as effective as any medication used to treat vulvar pain in vulvodynia [29].

Furthermore, when analyzing these therapeutic proposals one must ask what is their rationale. Is vulvodynia a problem of the pelvic floor muscles or a psychological disorder or the consequence of an unknown nociceptive source or is it secondary to a pathology of the vulvar mucosa or everything altogether? In reality, these proposed treatments are nothing more than a ‘trial and error’ approach as has been recognized by some authors [30].

And, fundamentally, it remains a mechanistic interpretation of pain when it is now clear that pain can be experienced without any tissue damage. Many patients report severe and disabling pain without evidence of disease or injury and, conversely, patients with severe pathological deterioration report few symptoms. In the last 20 years, there has been a real revolution in the understanding of the pain that has led to a paradigm shift. Significant factors in this change have been a greater understanding of the neurobiology of pain, the emergence of functional brain neuroimaging, and the assumption of the biopsychosocial model into clinical practice. We are witnessing the transition from a biomedical model, in which nociception and pain were considered almost synonymous terms, to a more complex, but more attractive vision, in which pain is interpreted as a response from the brain, and where nociception can play a highly variable role. As we have often emphasized, the problem of pain does not lie primarily in its treatment, but rather in the poor or inadequate vision of what pain essentially is.

If there is no radical change in the understanding of vulvodynia, women will continue to be subjected to unsuccessful examinations and treatments that will depend more on the biases and preferences of the health professional who treats them.

The consequences of vulvodynia in a woman’s life can be devastating due to its significant physical disability, limitation in activities of daily living, and psychological condition [31,32]. Vulvodynia has usually deleterious consequences on a couple’s sexuality and intimacy [33]. Therefore, this condition has more serious psychosocial consequences than other pain conditions. Many women report feelings of shame, worthlessness as a sexual partner, disaffection with their body, and low self-esteem [33,34,35]. Hence, vulvodynia is currently recognized as a major health problem [36].

One of the reasons for the considerable impact of vulvodynia on a woman’s life is that it is rarely understood by her spouse or partner, or by her closest relatives, nor by the doctors or health professionals to whom they consult. In the USA, a population-based study has shown that nearly 40% of women chose not to seek treatment, and of those who did, 60% saw 3 or more doctors, many of whom could not provide a diagnosis [4]. In Europe, only 10–25% of patients obtain the correct diagnosis from their first visit to the gynecologist and only 20% of gynecologists know the diagnosis and initiate appropriate treatment [24]. As has been noted, women with vulvodynia report many barriers to help-seeking [3].

In 2019, a consensus meeting was held between the International Society for the Study of Vulvovaginal Disease, the International Society for the Study of Women Sexual Health, and the International Pelvic Pain Society in which the descriptors of vulvodynia were divided into 4 groups: location, provocation, onset, and temporal pattern [37] (Table 3).

Depending on its location, a difference is made between localized vulvodynia (vestibulodynia, clitoridynia), generalized (the entire vulva, including the vestibule, clitoris, labia minora, and majora). Likewise, vulvodynia is classified according to whether the pain is stimulus-dependent or stimulus-independent, in provoked (insertional, contact), spontaneous or mixed vulvodynia. It also differentiates between primary provoked vulvodynia if symptoms started with the first time a provocative contact occurred, such as the insertion of a tampon or with the first sexual intercourse, while secondary provoked vulvodynia when the onset of symptoms is after a painless vulvar contact period. Regarding the temporal pattern, a difference is made between persistent, if the symptoms persist for at least 3 months (symptoms can be constant or intermittent), constant or intermittent, depending on whether they are always present or not, and finally between immediate or delayed, if the symptoms occur during physical contact or appear later.

The new classification of the *International Association for the Study of Pain* (IASP) classifies vulvodynia within the category ‘chronic primary visceral pain’ [38], which is based on the 2015 consensus terminology [12]. The 11th revision of the *International Statistical Classification of Diseases and Related Health Problems*, adopted by the WHO and that will come into effect force on 1 January 2022, has included vulvodynia within the chronic pain syndromes for the first time.

## 2. Pathophysiology of Vulvodynia

The traditional conceptualization of vulvodynia, like other chronic pain entities, has been clearly dualistic, either as a result of organic-physical mechanisms or psychological-sexual mechanisms. Despite the advances in the understanding of idiopathic pain and the recognition of neuroplastic changes as the cause of chronic and complex pain conditions, multiple pathophysiological mechanisms continue to be proposed in the development of vulvodynia, such as genetic factors, local inflammation mechanisms, hormonal deficiencies, peripheral neuropathic pain, pelvic floor muscle dysfunctions, etc. [3,39,40]. Recently it has been proposed an association between vulvodynia and the reported history of exposures to a number of household and work-related environmental toxins [41]. In the attached figure, we show the factors involved in vulvodynia (Figure 1).

### 2.1. Genetic Factors

Some studies suggest that some women have a genetic predisposition to suffer from vulvodynia. This predisposition could be related to genetic polymorphisms that increase the risk of candidiasis or other vulvar infections, to genetic changes that favor an exaggerated inflammatory response, or to changes that increase hormonal susceptibility to oral contraceptives, or some polymorphisms involved in the modulation of endogenous pain [42,43,44].

Genetics is an emerging discipline and, as often happens in these cases, perhaps too many answers in the field of chronic pain are expected. Multiple genes can be involved with genetic interactions, but also in genetic-environment interactions and epigenetic variants. Currently is difficult to make a reliable estimate of the genetic component of chronic pain conditions, because of the complex interactions between the genes and environment, psychological comorbidity, and aspects of family learning [45,46].

In some studies carried out in identical twins with CPP, a genetic component appears to be observed, but the contribution of this genetic predisposition does not reach a third of the total variation in the susceptibility to suffering from CPP [47,48]. Other studies have not found that the proposed genetic polymorphisms contribute to the development of vulvodynia [49,50].

### 2.2. Inflammatory/Infectious Factors

It has been hypothesized an inflammatory pathogenesis for vulvodynia [39,40], secondary to recurrent bacterial or candida infections [51] or, in some cases, following trauma to the vestibular mucosa [52]. It has been suggested that there is a relationship between natural killer cell number deficiencies and recurrent yeast infections. Thus, vulvodynia would express a central sensitization condition that persists after the resolution of the acute local inflammation [53].

Several groups have proposed the study of vaginal or plasma pro-inflammatory cytokine profiles as possible biomarkers of vulvodynia [54,55,56]. An increase in the number of mast cells in the vestibular tissue has been found in women with vulvodynia [57], as well as a systemic reduction in the number of *natural killer* cells compared to controls [58].

However, the results of these studies are inconsistent. A recent systematic review was carried out by Chalmers et al. [57] has concluded that current evidence is limited and contradictory regarding the presence of local and systemic inflammation in women with vulvodynia, including levels of cytokines, prostaglandin E2, T cells, B cells, mast cells, *natural killer* cells, and macrophages.

### 2.3. Hormonal Factors

Sexual responses, as well as genital pain, are modulated, in addition to neural pathways, by circulating levels of gonadal hormones [59]. It has thus been postulated that low estrogen levels could lead to vulvodynia and dyspareunia. The decline in estrogen levels can occur naturally or iatrogenically. The most common cause of low estrogen levels in women is menopause. Other natural causes include anovulation secondary to lactation, anorexia, hypothalamic amenorrhea, hyperprolactinemia, and excessive physical activity or physiological stress [60].

As iatrogenic causes, decreased circulating estrogen after oophorectomy and hysterectomy and combined hormonal contraceptive drugs are cited [60]. Combined hormonal contraceptives lead to a reduction in serum estradiol and free testosterone by decreasing ovarian production of estrogen and total testosterone. In addition, some combined hormonal contraceptives contain synthetic progestogens that act as antagonists of testosterone at the androgen receptor [60].

Combined hormonal contraceptives can cause changes in the vestibular mucosa, increasing its vulnerability to mechanical stress [61]. It has been suggested that the use of combined hormonal contraceptives before the age of 17 increases the relative risk of developing vulvodynia [62]. However, this association has not been found in population studies [63].

It has been shown that women with vulvar pain without an identifiable cause, but who started taking hormonal contraceptives may effectively be treated by discontinuing these contraceptives combined with the application of topical hormone therapy [64]. Based on this therapeutic combination, our group proposes to distinguish persistent vulvar pain due to a genitourinary syndrome from menopause due to lack of estrogens from vulvodynia (pain syndrome), which may appear overlapping in menopause, as proposed by the current terminology of Table 1 [65].

### 2.4. Peripheral Neuropathic Pain

Recent publications continue to sustain that vulvodynia is a neuropathic pain that in some cases is associated with a dysfunction of the pelvic floor muscles [27]. Without meeting the diagnostic criteria of the IASP, it is stated that vulvodynia is a neuropathic pain due to its burning nature and because of the hypersensitivity of the vulvar mucosa. This statement is a misinterpretation of what vulvodynia is. It should be remembered that if vulvar pain is secondary to entrapment or injury to a peripheral nerve, the term vulvodynia should no longer be used.

One of the proposals on the aetiology of vulvodynia, mainly provoked vulvodynia, is a greater nerve fiber proliferation in the vulvar vestibule. For example, some studies have found an increase in the density of C nociceptor endings [52,66]. In these cases, the proposed treatment is vestibulectomy [67,68,69]. However, on the one hand, it has been shown that an increase in the density of nociceptors is not consistently correlated with allodynia in the vestibular mucosa, with no significant differences being observed between those points that patients perceive as sensitive and those that are not [70].

### 2.5. Pelvic Floor Muscle Dysfunction

Vulvodynia has been associated with dysfunction of the pelvic floor muscles, such as hyperactivity, increased pelvic floor tone at rest, deficits in muscle control, and the presence of myofascial trigger points. [71,72]. The location of the pain is even related to the specific involvement of different muscles of the pelvic floor [73]. However, there is no evidence for these claims [74] and to accurately assess pelvic floor muscle dysfunction in clinical practice, other more objective methods such as 4D transperineal ultrasound or dynamometric speculum should be further developed [75].

It is not clear whether the observed muscle hypertonicity is causally related to the aetiology of vulvodynia or is the result of pain, given the cross-sectional designs of the studies to date [39,40]. As points out Micheletti et al. [76] the phenomenon of increased muscle tone of the pelvic floor muscles, frequently reported in women with provoked vulvodynia, is probably not the cause but the consequence of the pain. Women may also have spontaneous contraction of the pelvic floor muscles during attempted vaginal penetration. This pelvic floor muscle dysfunction may be the result of a protective reflex to prevent penetration or painful contact [40].

In conclusion, the relationship between genetic, inflammatory, infectious, hormonal, neuroproliferative, or muscular factors and vulvodynia has not been demonstrated. It should be remembered that the ISSVD classification clearly differentiates between vulvodynia and inflammatory or neuropathic pain secondary to infectious, inflammatory, neoplastic, or neurological disorders [1]. The IASP Interest Group on Neuropathic Pain (IASP NeuPSIG) states that the pain in vulvodynia should not be considered neuropathic and therefore it must fall into the category of dysfunctional pain [77].

### 2.6. Psychological Factors

The mechanistic dualism has led to understanding vulvodynia as a purely psychological disorder since women who suffer from it report more anxiety, fear of pain, hypervigilance, catastrophism, and depression [78]. The association between anxiety and sexual problems and pelvic pain has been suggested by several studies [33,79]. Psychosexual evaluation combined with psychotherapy is recommended as a therapeutic approach to vulvodynia [78,80].

However, pain is not a psychological phenomenon, but a real somatosensory experience generated by the CNS, in which psychosocial aspects have a great influence. Pain is a complex human experience in which many different factors interact, not only biological but also psychological, social, and cultural.

## 3. Vulvodynia and Neurobiological Approach

Vulvodynia is a clear example that a neurobiological approach is needed to understand chronic pain conditions without relevant nociception. The central sensitization paradigm led Woolf et al. [81]. The central sensitization paradigm led Woolf et al. to the development of a new classification of pain mechanisms, from a neurobiological perspective: nociceptive/inflammatory pain, neuropathic pain, and dysfunctional pain [81].

Dysfunctional pain is defined as pain that is not the consequence of a tissue injury, or a detectable inflammatory response or injury to the somatosensory system [82]. In Woolf’s words [83] ‘pain could in these circumstances become the equivalent of illusory perception, a sensation that has the exact quality of that evoked by a real noxious stimulus but which occurs in the absence of such an injurious stimulus. What characterizes dysfunctional pain is the presence of spontaneous or stimulus-dependent pain, sensory amplification, evoked by low and high-intensity stimuli and present with lack of stimulus.

Dysfunctional pain is therefore maladaptive pain, without any biological function, generated by the CNS, without stimulation of peripheral nociceptors [82]. Unlike inflammatory pain, which represents reversible pain hypersensitivity as an associated response to tissue inflammation, dysfunctional pain involves a maladaptive central sensitization since it does not have a protective function, like nociceptive pain, nor does it promote healing, like inflammatory pain.

As a consequence, the diagnosis of vulvodynia as dysfunctional pain should be based on the prior exclusion of specific disorders responsible for inflammatory and neuropathic pain, as recommended by the NeuPSIG guidelines in the assessment of neuropathic pain [84].

The first authors to propose this differentiation between nociceptive, inflammatory, neuropathic, and dysfunctional pain in vulvar pain were Micheletti et al. [85] in 2014. According to these authors [76] the problem is that many definitions of vulvodynia, including the one proposed by the ISSVD as “vulvar pain in the absence of relevant visible findings or a specific, clinically identifiable neurological disorder “, are completely ambiguous from a neurobiological perspective.

From a neurobiological perspective, vulvodynia should be regarded as dysfunctional pain. This has been defined by Micheletti et al. [85] as maladaptive, low threshold pain in the absence of peripheral tissue inflammation or neural damage, induced by exposure to acute physical or psychological precipitating events in the presence of an individual predisposition to produce or maintain abnormal central sensitization. Central sensitization involves abnormal long-term potentiation that can begin after physical precipitating events such as recurrent vulvovaginal candidiasis, lower urinary tract infections, or dermatologic pathology.

Changes derived from central sensitization such as hyperalgesia and allodynia have been demonstrated in vulvodynia, not only in the perineal area, but also in distant regions of the body [86,87]. Women with vulvodynia, for example, show increased pain responses after intradermal injection of capsaicin in the foot and forearm compared to control subjects [88]. Likewise, in patients with vulvodynia, an increased brain response has been observed with fMRI when pressing on different points of the vulva [89].

Vulvodynia is thus considered to be one of the “central sensitization syndromes”, a group of heterogeneous conditions that include fibromyalgia, chronic fatigue syndrome, irritable bowel syndrome, and temporomandibular joint disorder, among many others, characterized by symptoms such as pain and fatigue in the absence of clinically evident pathology [90,91].

### 3.1. Vulvodynia and Central Sensitization Pain

In recent years the term ‘central sensitization pain’ has become popular, similar in many aspects to dysfunctional pain [92,93,94,95,96,97]. Central sensitization pain is characterized by disproportionate pain, which implies that the severity of pain and perceived disability are disproportionate to the nature and extent of the injury or pathology [98].

A clinical algorithm has been developed to identify patients with central sensitization pain [98]. This algorithm includes aspects such as pain severity and its relationship with tissue disease or injury, pain distribution and hyperalgesia, sensory hypersensitivity, and some other clinical characteristics.

Levesque et al. [99] conducted a Delphi study with different international experts in pelvic pain proposing 10 criteria (*Convergences PP Criteria*) as a clinical tool to identify central sensitization in pelvic pain.

The problem with these proposals is that, since central sensitization would be the pathophysiological mechanism that could explain the transition to chronic pain, central sensitization is wrongly equated with chronic pain [100,101].

Recently, some doubts have arisen as to whether an increase in central sensitivity is relevant as an explanation for chronic pain. It has even been stated that the differences in pain thresholds may be more related to patient’s psychological distress and fear-avoidance [102,103].

Furthermore, while central sensitization may play an important role in the development of chronic pain, it is not yet clear whether, in some cases, these changes in central pain processing are primary or secondary to prolonged pain or emotional stress from pain or its interpretation [104]. Although many patients with chronic pain exhibit generalized hyperalgesia and allodynia, these characteristics can also be seen in inflammatory or nociceptive pain conditions, such as osteoarthritis or rheumatoid arthritis [105,106,107,108,109]. Therefore, central sensitization is not the differential feature of chronic pain conditions and does not play a major role in patients’ reporting of pain and disability [110].

### 3.2. Nociplastic Pain

In 2016 Kosek et al. [111] proposed the term nociplastic pain as a descriptor in which ‘pain arises from altered nociception despite no clear evidence of actual or threatened tissue damage causing the activation of peripheral nociceptors or evidence for disease or lesion of the somatosensory system causing the pain’. This new descriptor would include clinical situations in which there is no obvious activation of nociceptors or neuropathy, but in whom clinical and psychophysical findings suggest altered nociceptive function. Typical patient groups include those labeled as having fibromyalgia, complex regional pain syndrome type 1 (CRPS 1), other types of “musculoskeletal” pain (such as chronic “non-specific” low back pain, and “functional” visceral pain disorders (such as vulvodynia, irritable bowel syndrome, bladder pain syndrome, etc.) However, there is still much controversy about the validity of this new descriptor.

The problem with this descriptor, which claims to be ‘mechanistic’, is that it does not refer clearly to any mechanism. While nociceptive and neuropathic pain involves known mechanisms, nociplastic does not refer to any. This descriptor is based on the concept of ‘altered nociceptive function’, however, changes in the nociceptive system also occur in primary hyperalgesia (inflammatory pain), neuropathic pain, and when there is central sensitization.

Another problem is that this descriptor is reserved for patients who appear to have symptoms of neuropathic pain but who do not meet the criteria for its diagnosis. Therefore, this descriptor cannot be used in pain patients who do not show hypersensitivity, allodynia, etc. Therefore, it cannot be used in many patients with chronic pain without relevant nociception who do not have these symptoms. Once again it’s to keep looking for a mechanistic label for something humanly complex like chronic pain.

### 3.3. Vulvodynia and Complex Pain

Pain is suffered by a person. Understanding the patient’s pain cannot be reduced to the identification of pain mechanisms (nociceptive, neuropathic, nociplastic), forgetting the person. A complex human experience cannot be reduced to a mechanism. To understand the clinical situation of the patient, we cannot continue with the body-mind dichotomy. Health professionals have mostly tried to explain the patient’s clinical situation from a biological perspective, trying to differentiate which pain mechanisms are involved and trying to identify biomarkers. This is a biological bias that feels uncomfortable in front of psychological and social factors, always considering them as secondary phenomena to pain.

Pain is experienced by a ‘whole’ person and the characterization of a patient’s pain should include all aspects involved in this experience: biological, psychological, and social. We, therefore, suggest the use of the term *complex pain* as it is more encompassing and better characterizes the experience of the patient with chronic pain.

Complex pain can be defined as pain unrelated to any peripheral nociceptive input or injury or disease of the somatosensory system, associated with some degree of central hyperexcitability, significant emotional distress, disability, and with a significant impact on the individual’s work and social aspects.

Vulvodynia is a complex pain, is essentially the same as many other chronic pain conditions, such as fibromyalgia, irritable bowel syndrome, atypical facial pain, urethral/bladder pain syndrome, and tension-type headache, in which there is hypersensitivity to pain but no noxious stimulus, no inflammation, and no structural damage to the somatosensory system.

A relevant aspect is that complex pain shares some characteristics that were previously attributed to neuropathic pain, such as temporal summation in response to repeated stimuli (wind up), the pain of diffuse distribution, and reduced pain thresholds. A relevant proportion of patients with complex pain show high scores in some neuropathic pain scoring tools [112]. These common features with neuropathic pain, such as spontaneous electrical pain, mechanical hyperalgesia, thermal and mechanical allodynia, temporal summation, and somatosensory perceptual abnormalities, are associated with central sensitization phenomena that may also be present in complex pain. These neuropathic-type symptoms have confused clinicians and researchers who have assumed that there is a neuropathic component in all chronic pain and that clinical conditions in which there is no obvious nociceptive source are central neuropathic pain.

It should be emphasized that the degree of central sensitization in patients with vulvodynia can be highly variable, just as it is in inflammatory and neuropathic vulvar pain. In addition, complex pain can also present as acute pain.

The distribution of pain is often not neuroanatomically plausible with a pathology of a somatic or visceral structure or with peripheral neuropathic pain. Also, patients report not only vulvar pain but symptoms that seem to be related to different systems such as urologic, proctologic, or musculoskeletal systems. As an example, patients with vulvodynia frequently report symptoms such as tenesmus or dysuria that are associated with urinary tract infection when it is confirmed that such infection is not present.

The comorbidity of vulvodynia with various chronic pelvic pain syndromes, as well as with other complex pain conditions such as fibromyalgia, is tremendously high [14,90,113]. Many women with vulvodynia also report previous premorbid conditions. Reed et al. [114] conducted a longitudinal study among 1037 women and analyzed risk factors for vulvodynia. These authors found that in addition to postcoital pain, already suggested in previous publications, the most relevant risk factor was having a premorbid history of nonspecific urogenital symptoms.

### 3.4. Complex Pain Identifiers

In patients with complex pain, different aspects dominate the clinical picture and mediate treatment outcomes, such as central hyperexcitability, illness behaviors, psychological distress, and disability.

The author has recently published a complex pain identification system based on a series of identifiers [115]. The objective of these is to promote the recognition of complex chronic pain not associated with a nociceptive input or when it is not relevant, including both the pain characteristics and different associated behavioral and psychosocial aspects (Table 4).

The triggers of complex pain can be both biological, such as a previous severe or prolonged nociceptive or neuropathic pain, as well as psychosocial such as generalized anxiety, etc. Abnormal central pain processing can also be triggered by psychological, sexual, and/or social factors such as adverse childhood experiences, family learning, traumatic experiences such as sexual abuse, anxiety, and depression [116,117,118]. Vulvodynia has been found to be four times more likely in women with a history of depression and anxiety [20].

## 4. Conclusions

Vulvodynia is a complex pain disorder whose treatment is usually a failure from a narrow biomedical perspective. Understanding the patient’s pain problem requires determining whether there is an actual injury or disease or it is a dysfunctional response of the Central Nervous System. Thus, it is necessary to differentiate the type of chronic pain from the mechanisms involved (nociceptive, inflammatory, and neuropathic). In addition, a new category, complex pain, needs to be included for people with chronic pain (but also acute pain) for whom there is no relevant nociceptive information.

An adequate taxonomy of pain, which implies a better understanding of pain mechanisms, has the potential to improve clinical decision-making and has important consequences both for the diagnosis and for the treatment and prognosis of patients with pain [119]. As Baranowski points out [120], ‘It is the responsibility of all involved in the management of patients who have chronic pain as a symptom to review the significant recent research into chronic pain mechanisms and reconsider their management approach to this disease condition. This is particularly true for what is known as the chronic pelvic pain syndromes, that is, pain perceived to be related to the pelvis where there are no well-recognized pathologies’.

Despite the different hypotheses about the aetiology of vulvodynia that have been proposed, there is currently no evidence for any of them. Vulvodynia is a clear example of pain not associated with relevant nociception, which, as in other chronic pain conditions, determines high emotional distress, as well as significant disability. That is why vulvodynia must be viewed from a neurobiological perspective that also incorporates the different psychosocial aspects frequently involved. Ten identifiers of vulvodynia have been proposed that cover both pain characteristics and associated psychosocial aspects.

If the patient’s vulvar pain is secondary to a nociceptive source or an injury of the somatosensory system, strategies that treat both the source of pain and those aimed at modulating the CNS response, either with pharmacological or interventional treatment, are justified. However, if vulvar pain is related to complex pain, as is the case of vulvodynia, strategies to modulate altered central pain processing is necessary, changing the patient’s erroneous cognitions about their pain, and also reducing fear avoidance-behaviors and the disability of the patient. As it is emphasized in one of the definitions of pain proposed by the IASP, adequate pain treatment requires understanding it as a complex human experience with multiple dimensions: somatosensory, emotional, cognitive, and social [121]. For this reason, currently, numerous evidence-based clinical guidelines establish that chronic pain must be approached with a multidisciplinary approach that incorporates adequate patient education on their clinical condition [35,122,123,124,125,126].

This can be achieved by explaining to the patient the mechanism involved in this altered pain processing using pain neurobiology. Education in neurobiology or neuroscience of pain is an emerging strategy in the management of patients with chronic pain. This educational model started by Butler and Moseley about 20 years ago [127] is a promising strategy in the management of chronic pain, as has been stated in the latest systematic review on this therapeutic approach [128]. Pain neuroscience education is justified because all pain has a meaning. The patient’s attribution of the meaning of the symptoms, in particular their perception of pain as a sign of a serious problem, their beliefs about the potential impact on their lives and their future, are critical in the perception of pain [129,130]. Education is the cornerstone of our approach, with the premise that the better an individual understands their condition; the more able they are to change their maladaptive perceptions of pain. [128]. The second strategy is based on the use of desensitization and gradual exposure techniques. Pain neuroscience education is thus capable of generating a conceptual change in patients with complex chronic pain, allowing them to understand that pain, no matter how intense, does not imply any damage and that it is a reversible condition thanks to the plasticity of the CNS. But this change in perceptual inference has also been favored by facilitating exposure to those specific pain-triggering stimuli and thus allowing progressive desensitization. The authors of this article are currently conducting some clinical trials for the treatment of vulvodynia. In a recently completed study (pending publication) in women with vulvodynia and dyspareunia, it has been shown that these proposed strategies are capable of significantly eliminating pain and disability. In addition, in this study with fMRI techniques, it has been shown that the treatment produces significant changes in brain connectivity. We hypothesize that the effects obtained with our intervention are based on the ability of our brain to generate a new pain-free perceptual inference from a change in the understanding of vulvodynia.

We hope our proposal will facilitate the acknowledgment and recognition that vulvodynia is a ‘real pain’ but no related to any tissue injury or disease. We also hope that this new understanding will lead to further investigation into vulvodynia and associated CNS changes and will open new avenues in the treatment of women with this debilitating condition.

## Figures and Tables

**Figure 1 ijerph-18-06639-f001:**
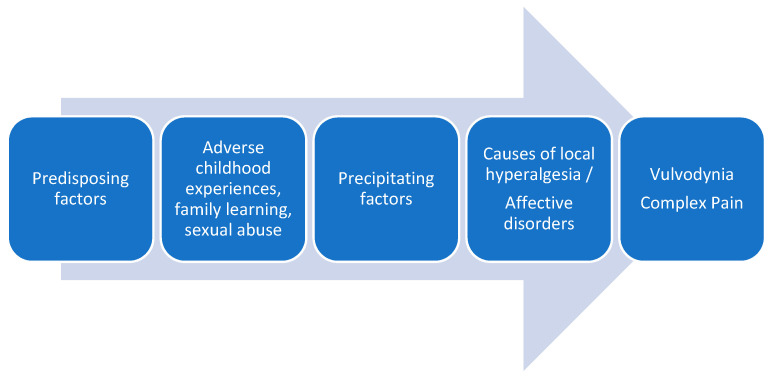
Factors involved in vulvodynia.

**Table 1 ijerph-18-06639-t001:** 2015 Consensus Terminology and Classification of Persistent Vulvar Pain and Vulvodynia (ISSVD, ISSWSH, and IPPS) [11].

Vulvar Pain Caused by a Specific Disorder	Vulvodynia
**Infectious** (e.g., recurrent candidiasis, herpes)	Vulvar pain of at least 3 months’ duration, without clear identifiable cause, which may have potential associated factors
**Inflammatory** (e.g., lichen sclerosus, lichen planus, immunobullous disorders)	**Localized** (e.g., vestibulodynia, clitorodynia) **Generalized** **Mixed** (localized and generalized)
**Neoplastic** (e.g., Paget disease, squamous cell carcinoma)	
**Neurologic** (e.g., postherpetic neuralgia, nerve compression, or injury, neuroma)	**Provoked** (e.g., insertional, contact) **Spontaneous** **Mixed** (provoked and spontaneous)
**Trauma** (e.g., female genital cutting, obstetrical)	
**Iatrogenic** (e.g., postoperative, chemotherapy, radiation)	**Onset** (primary or secondary)
**Hormonal deficiencies** (e.g., genitourinary syndrome of menopause, lactational amenorrhea)	**Temporal pattern** (intermittent, persistent, constant, immediate, delayed

**Table 2 ijerph-18-06639-t002:** 2015 ISSVD, ISSWSH, and IPPS Consensus Terminology and Classification of Persistent Vulvar Pain and Vulvodynia Vulvodynia—Potential Associated Factors [12].

Potential Factors Associated with Vulvodynia
Comorbidities and other pain syndromes (e.g., painful bladder syndrome, fibromyalgia, irritable bowel syndrome, temporomandibular disorder; level of evidence 2) Genetics (level of evidence 2) Hormonal factors (e.g., pharmacologically induced; level of evidence 2) Inflammation (level of evidence 2) Musculoskeletal (e.g., pelvic muscle overactivity, myofascial, biomechanical; level of evidence 2) Neurologic mechanisms ○ Central (spine, brain; level of evidence 2) ○ Peripheral: neuroproliferation (level of evidence 2) Psychosocial factors (e.g., mood, interpersonal, coping, role, sexual function; level of evidence 2) Structural defects (e.g., perineal descent; level of evidence 3)

**Table 3 ijerph-18-06639-t003:** Vulvodynia Descriptors (ISSVD, ISSWSH, and IPPS) [37].

Definitions of Vulvodynia Descriptors (ISSVD, ISSWSH, and IPPS) [37]		
Descriptor		Definition
**Location**	Localized	Involvement of a portion of the vulva.
	Generalized	Involvement of the whole vulva.
**Provocation**	Provoked	The discomfort is provoked by physical contact.
	Spontaneous	The symptoms occur without any provoking physical contact
**Onset**	Primary	Onset of the symptoms occurs with first provoking physical contact
	Secondary	Onset of the symptoms did not occur with first provoking physical contact
**Temporal pattern**	Persistent	The condition persists over a period of at least 3 months
	Constant	The symptoms are always present
	Intermittent	The symptoms are not always present
	Immediate	The symptoms occur during the provoking physical contact
	Delayed	The symptoms occur after the provoking physical contact

**Table 4 ijerph-18-06639-t004:** Characteristics of complex pain [115].

Complex Pain Features [115]
1. There is no history of a relevant injury or disease in the vulva
2. The nature of pain, location, severity of pain, and duration is inconsistent with a secondary vulvar pain
3. Frequently widespread pain distribution, allodynia, and hyperalgesia
4. No physical o medical precipitant factors
5. Comorbidity: frequent association of urological, coloproctological, abdominal, and musculoskeletal symptoms
6. Disproportionate disability
7. Generalized hypersensitivity to many stimuli: bright light, cold/heat, sound/noise, weather, stress, food
8. Abnormal therapeutic response
9. Associated cognitive, emotional, and behavioral factors
10. Associated psychosocial aspects

## Data Availability

Not applicable.

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
