# Peer review of "Vulvodynia—It Is Time to Accept a New Understanding from a Neurobiological Perspective"

_ijerph, 2021, doi:10.3390/ijerph18126639_

Round 1

Reviewer 1 Report

First of all, this article underscores how important it is to explain to patients with chronic vulvar pain that although sometimes we cannot point out where their pain comes from physically or anatomically, it is real. The article stresses how crucial it is to listen to patients and believe their stories. It also reminds the reader that this pathology is frequent and debilitating. This alone is necessary.

The first part of the article relating the history of the terminology of vulvodynia is quite interesting and helps the comprehension of the authors’ critique of the ‘trial and error’ way of treating vulvodynia. However, some information is redundant between the tables and the text, especially lines 162-173.

The authors point of view on the taxonomy of vulvodynia is valid and interesting, but the third part of the article explaining this (starting line 297), needs some work. Long explanations of afterward invalidated theories are misleading (lines 334-412). In particular, lines 356-368 do not help to understand Table 4. A shorter explanation of the algorithm without showing it as a table would suffice. Again, Table 5 is shown but its proposal is invalidated in the next paragraph (lines 374-388). It would be clearer to the reader if the table was not shown.

Lines 413 and onwards describe the main proposition of the authors and the ideas are clearly explained. It would be interesting for the authors to develop on how to address vulvodynia from the complex pain point of view (so lightly addressed in final lines 508-511)

Finally, this article is contributory to the field. The proposition on dysfunctional pain/complex pain submitted here is pertinent.

Author Response

Response to Reviewer 1 Comments

Many that’s for your comments

First of all, this article underscores how important it is to explain to patients with chronic vulvar pain that although sometimes we cannot point out where their pain comes from physically or anatomically, it is real. The article stresses how crucial it is to listen to patients and believe their stories. It also reminds the reader that this pathology is frequent and debilitating. This alone is necessary.

Point 1: The first part of the article relating the history of the terminology of vulvodynia is quite interesting and helps the comprehension of the authors’ critique of the ‘trial and error’ way of treating vulvodynia. However, some information is redundant between the tables and the text, especially lines 162-173.

Response 1: As the reviewer points out the text in these lines (162-173) maybe redundant as there is an explanation of the table 3. The text in the table has been reduced.

Point 2: The authors point of view on the taxonomy of vulvodynia is valid and interesting, but the third part of the article explaining this (starting line 297), needs some work. Long explanations of afterward invalidated theories are misleading (lines 334-412). In particular, lines 356-368 do not help to understand Table 4. A shorter explanation of the algorithm without showing it as a table would suffice. Again, Table 5 is shown but its proposal is invalidated in the next paragraph (lines 374-388). It would be clearer to the reader if the table was not shown.

Response 2: The explanation of the Algorithm for the classification of central sensitization and of the Convergences PP Criteria have been reduced and the tables 4 and 5 have been deleted

Point 3: Lines 413 and onwards describe the main proposition of the authors and the ideas are clearly explained. It would be interesting for the authors to develop on how to address vulvodynia from the complex pain point of view (so lightly addressed in final lines 508-511)

Response 3: We have included a new paragraph to explain how we address the treatment of vulvodynia:

As it is emphasizes in one of the definitions of pain proposed by the IASP, adequate pain treatment requires understanding it as a complex human experience with multiple dimensions: somatosensory, emotional, cognitive and social [1].

For this reason, currently numerous evidence-based clinical guidelines establish that chronic pain must be approached with a multidisciplinary approach that incorporates adequate patient education on their clinical condition [2] [3-7].

This can be achieved by explaining to the patient the mechanism involved in this altered pain processing using pain neurobiology. Education in neurobiology or neuroscience of pain is an emerging strategy in the management of patients with chronic pain. This educational model started by Butler and Moseley about 20 years ago [8] is a promising strategy in the management of chronic pain, as has been stated in the latest systematic review on this therapeutic approach [9].

Pain neuroscience education is justified because all pain refers to a meaning. The patient's attribution of the meaning of the symptoms, in particular their perception of pain as a sign of a serious problem, their beliefs about the potential impact on their lives and their future, are critical in the perception of pain [10] [11].

Education is the cornerstone of our approach, with the premise that the better an individual understands their condition; the more able they are to change their maladaptive perceptions of pain. [9]. The second strategy is based on the use of desensitization and gradual exposure techniques.

Pain neuroscience education is thus capable of generating a conceptual change in patients with complex chronic pain, allowing them to understand that pain, no matter how intense, does not imply any damage and that it is a reversible condition thanks to the plasticity of the CNS. But this change in perceptual inference has also been favored by facilitating exposure to those specific pain-triggering stimuli and thus allowing a progressive desensitization. The authors of this article are currently conducting some clinical trials for the treatment of vulvodynia. In a recently completed study (pending publication) in women with vulvodynia and dyspareunia, it has been shown that these proposed strategies are capable of significantly eliminating pain and disability. In addition, in this study with fMRI techniques, it has been shown that the treatment produces significant changes in brain connectivity. Our hypothesis is that the effects obtained with our intervention are based on the ability of our brain to generate a new pain-free perceptual inference from a change in the understanding of vulvodynia.

We hope our proposal will facilitate the acknowledgment and recognition that vulvodynia is a ‘real pain’ but no related to any tissue injury or disease and. We also hope that this new understanding will lead to further investigation into vulvodynia and associated CNS changes and will open new avenues in the treatment of women with this debilitating condition.

Finally, this article is contributory to the field. The proposition on dysfunctional pain/complex pain submitted here is pertinent.

Reviewer 2 Report

The manuscript titled “Vulvodynia. It is time to accept a new understanding from a  neurobiological perspective analyses vulvodynia from a neurobiological perspective. Different clinical identifiers of vulvodynia are presented from a neurobiological and psychosocial perspective. This study is original and useful. Topic is interesting enough to attract the readers’ attention. This work could be published after major revision.

My observations are as follows:

  1. Check that all the tables are in the style of the journal

  1. Tables 1 and 2 are not very clear. I would make them clearer. They are a bit confusing.

  1. I would also cite references for each aspect considered. In other words : the data of Table 1 derives all from reference 12? if they do not all derive from reference 12, it would be better to specify each aspect to which reference it does.

  1. I suggest to add images to summarize the mechanisms involved instead of some tables and to give some colour to this work. This attracts the reader

  1. The quality of Table 4 needs to be improved.

  1. English revision of the entire manuscript is necessary

  1. The table 5 also needs improvements. It is a bit confusing.

  1. Explain the contents of the tables in the text

  1. I would add further details about association between vulvodynia and the reported history of exposures to a number of household and work-related environmental toxins. (authors may refer to: PMID: 30307787) since studies find link between air pollution and neurological disorders

Author Response

Response to Reviewer 2 Comments

Many thanks for your comments

Point 1: Check that all the tables are in the style of the journal.

Response 1: I have check them and they seem to be in the style of the journal

Point 2: Tables 1 and 2 are not very clear. I would make them clearer. They are a bit confusing.

Response 4:  these tables are the same of the paper referenced.

Point 3: I would also cite references for each aspect considered. In other words : the data of Table 1 derives all from reference 12? if they do not all derive from reference 12, it would be better to specify each aspect to which reference it does.

Response 3:  the data of table 1 and 2 derive all from the reference 12.

Point 4: I suggest to add images to summarize the mechanisms involved instead of some tables and to give some colour to this work. This attracts the reader

Response 4:  

Point 5: The quality of Table 4 needs to be improved.

Response 5:  Table 4 has been deleted

Point 6: English revision of the entire manuscript is necessary

It will be sent to the journal to be revised

Response 6: It is going to be sent.

Point 7:The table 5 also needs improvements. It is a bit confusing.

Response 7:  Table 4 has been deleted

Point 8: Explain the contents of the tables in the text

Response 8:  We have already done it

Point 9: I would add further details about association between vulvodynia and the reported history of exposures to a number of household and work-related environmental toxins. (authors may refer to: PMID: 30307787) since studies find link between air pollution and neurological disorders

Response 9:  Thanks for your recommendation. Certainly, we do not have now a full knowledge all the associated factors in vulvodynia such as in many complex pain conditions. 

This paper about the association between vulvodynia and the reported history of exposures to a number of household and work-related environmental toxins has been commented in the text.

‘Recently it has been proposed an association between vulvodynia and the reported history of exposures to a number of household and work-related environmental toxins [41]’

Round 2

Reviewer 2 Report

Accept in present form